# Catatonia in a Possible Case of Moderate Neuroleptic Malignant Syndrome: A Case Report

**DOI:** 10.3390/reports8030134

**Published:** 2025-08-02

**Authors:** Daniel Ungureanu, Patricia-Ștefania Mitrea, Silvina Iluț, Aurora Taloș, Cătălina-Angela Crișan

**Affiliations:** 1Department Pharmacy I, Discipline of Pharmaceutical Chemistry, “Iuliu Hațieganu” University of Medicine and Pharmacy, 41 Victor Babeș Street, 400012 Cluj-Napoca, Romania; daniel.ungureanu@elearn.umfcluj.ro; 2“Prof. Dr. Ion Chiricuță” Oncology Institute, 34-36 Republicii Street, 400015 Cluj-Napoca, Romania; 3Department Pharmacy II, Discipline of Clinical Pharmacy, “Iuliu Hațieganu” University of Medicine and Pharmacy, 12 Ion Creangă Street, 400010 Cluj-Napoca, Romania; 4Clinical Hospital of Infectious Diseases, 23 Iuliu Moldovan Street, 400003 Cluj-Napoca, Romania; patricia.stef.mitrea@elearn.umfcluj.ro; 5Department of Neurosciences, Faculty of Nursing and Health Sciences, “Iuliu Hațieganu” University of Medicine and Pharmacy, 8 Victor Babeș Street, 400012 Cluj-Napoca, Romania; 6Department of Neurosciences, Faculty of Medicine, “Iuliu Hațieganu” University of Medicine and Pharmacy, 8 Victor Babeș Street, 400012 Cluj-Napoca, Romania; talos_aurora_elena@elearn.umfcluj.ro; 7Department of Neurosciences, Discipline of Psychiatry and Pediatric Psychiatry, “Iuliu Hațieganu” University of Medicine and Pharmacy, 43 Victor Babeș Street, 400012 Cluj-Napoca, Romania; ccrisan@umfcluj.ro; 8First Psychiatric Clinic, Cluj County Emergency Clinical Hospital, 43 Victor Babeș Street, 400012 Cluj-Napoca, Romania

**Keywords:** catatonia, neuroleptic malignant syndrome, antipsychotics, benzodiazepines, autoimmune encephalitis

## Abstract

**Background and Clinical Significance**: Neuroleptic malignant syndrome (NMS) is a life-threatening condition usually caused by the exposure to antipsychotics. This case report presents a catatonia syndrome that may have developed in the context of a moderate NMS. **Case Presentation**: An 18-year-old male patient presented with a treatment-resistant catatonia syndrome that debuted 2 weeks prior to the presentation (creatin kinase levels = 4908 U/L, maximum temperature = 38.9°C, white blood count = 13.20 × 10^9^/L, Bush–Francis Catatonia Rating Scale = 30 points). Possible organic causes of catatonia were ruled out, according to the negative results obtained. The patient’s condition improved under benzodiazepine treatment and he was later discharged. After discharge, the catatonia was attributed to a possible NMS with moderate severity. The diagnosis was supported by NMS Diagnosis Criteria Score = 85 points and the presence of Levenson’s triad. **Conclusions**: This case highlights the concomitant manifestation of both catatonia and NMS in the same patient and the difficulty of establishing a correct diagnosis involving both entities.

## 1. Introduction and Clinical Significance

Catatonia is a severe psychomotor syndrome that can be present in both psychiatric disorders and other medical conditions, such as infections, neurological and endocrine conditions, and medication and drug withdrawals. It was previously thought that catatonia can only be associated with schizophrenia, though nowadays it is a standalone nosological entity described in both the *Diagnostic and Statistical Manual of Mental Disorders, Fifth Edition, Text Revision* (DSM-5-TR) and the *International Classification of Diseases, Eleventh Revision* (ICD-11) [1,2,3,4,5].

Psychiatric catatonia represents an exclusion diagnosis, especially in new patients, and it is only confirmed when other organic causes have been ruled out. The diagnosis of catatonia disorder due to general medical conditions should not be established when catatonia is in the context of delirium or neuroleptic malignant syndrome (NMS) [3,6,7,8].

According to Fink and Taylor [9], catatonia nowadays is mostly recognized in children and adolescents, including those with autism and intellectual disabilities.

NMS is a serious and life-threatening iatrogenic syndrome characterized by muscular rigidity, hyperthermia, loss of consciousness, and autonomous instability. The syndrome appears usually in the context of exposure to dopaminergic antagonists, such as antipsychotics, or the abrupt discontinuation of dopaminergic agonists, such as antiparkinsonian drugs. Other medications include metoclopramide, lithium, valproic acid, tetrabenazine, and antidepressants [10,11,12]. Drug–drug interactions have been associated with an increased risk of NMS. According to a study by Kyotani et al. [13], the concomitant administration of risperidone/lithium, clomipramine/risperidone, and olanzapine/aripiprazole combinations had the highest risks of NMS. This strengthens the fact that NMS is not necessarily limited to typical antipsychotics [13].

Other risk factors for NMS include male gender and younger age [14]. According to the DSM-5-TR, the incidence of NMS in individuals with antipsychotic treatment is around 0.01–0.02% [12].

Debates exist whether NMS is a form of malignant catatonia, described as a type of catatonia with clinically significant autonomous abnormalities. Historically, both entities were considered indistinguishable, though a potential differential diagnosis between the two can be based on the recent history of dopaminergic antagonist intake, the type of muscular rigidity, “lead pipe” in NMS and waxy in malignant catatonia. The presence of prodrome and the absence of laboratory abnormalities are particular to malignant catatonia [8,9,15,16,17]. Attempts to differentiate between NMS and catatonia represent a point of interest in the literature [18,19], highlighting the difficulty in separating and diagnosing them properly.

According to the Deutsche Gesellschaft für Psychiatrie und Psychotherapie, Psychosomatik und Nervenheilkunde e. V. (DGPPN) S3 guideline for schizophrenia [20], there are several stages of NMS, depending on the severity of symptoms and clinical presentation, as presented in Table 1 [20,21,22,23]. This guideline considers catatonia as a clinical symptom of NMS.

The concept of drug-induced catatonia has been previously reported in the literature [23,24]. The term neuroleptic-induced catatonia (NIC) has been used to describe the extrapyramidal effects associated with catatonia after neuroleptic administration. A study conducted by Lee concluded that NIC and NMS are disorders on the same spectrum and revealed no indication that extrapyramidal symptoms may degenerate into NIC [24].

The management of catatonia and NMS requires intensive care and prolonged hospitalization, thus providing an increased burden for both the medical staff and caregivers [25].

This report aims to present a clinical case of catatonia that may have developed in the context of moderate NMS in a patient with a recent history of antipsychotic intake.

## 2. Case Presentation

### 2.1. Patient Information

An 18-year-old male patient, with a documented history of psychiatric illness, was transferred on 28 February 2025, from the “Dr. Constantin Opriș” County Emergency Hospital in Baia Mare, Romania, to the First Psychiatric Clinic inside Cluj County Emergency Clinical Hospital in Cluj-Napoca, Romania, for a treatment-resistant catatonia syndrome that debuted 2 weeks prior to the admission. On presentation, the patient was in a normal psychomotor state, stuporous, with marked hypertonia and waxy flexibility. The patient was nasogastric fed and urinary catheterized.

Heteroanamnesis revealed that the chief complaint debuted 2 weeks prior to the admission to the “Dr. Constantin Opriș” County Emergency Hospital in Baia Mare, Romania. The patient started to present persecutory and prejudice paranoid delusions, as well as psychomotor restlessness along with episodes of psychomotor inhibition, marked generalized anxiety, hypoprosexia, hypomnesia, bradypsychia, and mixed insomnias. The mother of the patient described that he had bizarre behavior and, later, he developed social withdrawal, with selective mutism (he talked only with his parents), and was afraid of being alone. Ultimately, he developed muscular rigidity and communicated only through stereotypies. The chief complaint degenerated into stuporous catatonia, for which he was admitted to hospital. After one week in the hospital with stagnant evolution, the patient was transferred to the First Psychiatric Clinic inside Cluj County Emergency Clinical Hospital in Cluj-Napoca, Romania.

According to the patient information, there was no relevant family history. The psychiatric medical history of the patient included mild intellectual disability and anxious–depressive disorder, while the non-psychiatric medical history included cardiorespiratory arrest after birth, vitamin D deficiency, and stasis bronchitis (diagnosed in late February 2025). The patient had no known allergies and/or intolerances.

The patient lived with his parents and was studying at a vocational school. Previously, he had to repeat the fifth grade.

There was no history of alcohol or illicit substance use. Prior to the admission at the “Dr. Constantin Opriș” County Emergency Hospital in Baia Mare, Romania, the patient was adherent to a psychotropic treatment consisting of trazodone 150 mg/day p.o., gabapentin 100 mg/day p.o., and piracetam 400 mg/day p.o. During the admission in Baia Mare, the patient initially received diazepam 10 mg i.m. and valproic acid 300 mg p.o., with brief symptomatic improvement. Then, when the symptoms evolved towards catatonia, the treatment consisted of clonazepam 1 mg/day p.o., titrated up to 6 mg/day p.o., haloperidol 5 mg/day i.m., switched to 2.5 mg/day p.o., trihexyphenidyl 6 mg/day p.o., aripiprazole 10 mg/day p.o., and zolpidem 10 mg/day p.o. The last one was administered only if insomnias were present. A timeline of the evolution of the psychotropic treatment during the admission in Baia Mare is represented in Figure 1.

The patient was transferred to Cluj-Napoca with the following psychotropic treatment: valproic acid 500 mg/day p.o., aripiprazole 10 mg/day p.o., and zolpidem 10 mg/day p.o. Similarly, zolpidem was administered only during insomnias. The non-psychotropic treatment consisted of ceftriaxone 4 mg/day i.v., for 10 days, dexamethasone 4 mg/day i.v., for 10 days, enoxaparin 400 IU/day s.c., pantoprazole 40 mg/day p.o., and a probiotic food supplement, for 10 days.

### 2.2. Mental State and Physical Examinations

Mental state examination on admission revealed a patient with good hygiene, kempt appearance, and appropriate clothing. Expressive language was abolished due to the catatonic state, mutism, and suspicious attitude. Non-verbal communication, such as gestures and facial expressions, was also abolished. Abnormal postures, muscular rigidity, and catalepsy were present, leading to overall bizarre behavior.

Cognitive assessment was markedly impaired due to the catatonic state. The patient had impaired attention, with distractibility and difficulties in maintaining focus (global hypoprosexia), and was disoriented to time and place.

The patient experienced complex auditory and visual hallucinations, which were revealed through heteroanamnesis. The thought content was dominated by persecutory and prejudice delusions, leading to fear. Poverty of thought and affective anesthesia were also present.

The patient displayed a general lack of motivation (global hypobulia) and social withdrawal. Basic instincts, such as self-preservation and feeding, were diminished. Nocturnal rhythm was impaired, leading to mixed insomnias. Insight could not be assessed.

The physical examination revealed muscular rigidity, marked hypertonia, inability to stand still or to walk, and crackles during pulmonary examination. Neurological examinations revealed generalized tremors, no signs of meningeal irritation, and positive bilateral Babinski sign. The examination of the other organs was unremarkable.

Psychometric evaluation yielded the following results: Bush–Francis Catatonia Rating Scale (BFCRS) = 30 out of 69 points (moderate severity), Positive and Negative Syndrome Scale (PANSS) = 127 points (severely ill—20 points on the Positive Scale, 40 points on the Negative Scale, and 67 points on the General Psychopathology Scale), Clinical Global Impression-Severity (CGI-S) = 5 out of 7 points (markedly ill), and Clinical Global Impressions Scale-Improvement (CGI-I) = 2 out of 7 points (not much improved) [26,27,28,29]. The English versions of the BFCRS and PANSS scales were used, which were verbally translated in Romanian for the patient. The scales were applied by non-certified raters.

### 2.3. Clinical Findings and Diagnosis

Vital signs throughout the admission ranged as follows: blood pressures between 86/49 mmHg and 113/78 mmHg, heart rates between 78 bpm and 106 bpm, peripheral oxygen saturations (SpO_2_) between 97% and 98%, and temperatures between 36.5 °C and 38.9 °C. The temperature was over 38.0 °C on two different occasions, initially 38.9 °C, then 38.2 °C. No abnormal fluctuations of the blood pressure, heart rate, and SpO_2_ were observed during a time span of 24 h.

Routine tests evidenced hepatic cytolysis syndrome, rhabdomyolysis syndrome, non-specific inflammatory syndrome, infection, folate deficiency, and hypoglycemia. The results are summarized in Table 2.

In order to establish the etiology of the catatonia, several investigations were performed to rule out organic causes (infections, vascular, neurological, autoimmune, endocrine, or drug-induced causes).

The initial investigations were performed to rule out infectious causes, in particular, meningitis and encephalitis, and vascular causes. A native cranial CT scan and a cerebral MRI scan with contrast (Figure 2), performed at the “Dr. Constantin Opriș” County Emergency Hospital in Baia Mare, Romania, revealed no pathological modifications (Table 3). Additionally, anti-*Borellia burgoderferi* (sensu lato) IgG and IgM antibodies, along with human immunodeficiency virus (HIV) antibodies and p24 antigen, yielded negative results.

Several new tests from the blood and cerebrospinal fluid (CSF) samples were performed in Cluj-Napoca, to detect pathogens or antibodies related to an infection. The results were negative in all instances, except for anti-Varicella zoster virus (VZV) IgG antibodies, indicating a previous exposure to this virus. The full panel of investigations for infectious causes is summarized in Table 4, Table 5 and Table 6.

In order to rule out neurological causes, such as epilepsy, autoimmune diseases like anti-NMDA receptor encephalitis (a type of autoimmune encephalitis), prion diseases, and paraneoplastic neurological syndromes, several antibodies were tested from the blood and CSF samples. An electroencephalogram (EEG) was also performed. The EEG revealed generalized slowing without abnormal graphic elements and the results for all antibody tests were negative. The full panel of investigations for neurological causes is summarized in Table 7.

In order to rule out endocrine causes, especially thyroid disfunctions, the serum levels of thyroid hormones (FT4 and TSH) were also determined. The results showed that both hormones were in normal ranges (Table 2), thus excluding potential thyrotoxicosis.

An abdomen and pelvis CT scan with contrast was performed, following the indications from a general surgical examination, after no bowel transit was present for 7 days. No pathological modifications were evidenced.

Before discharging, the patient developed a urinary tract infection (UTI) with *Klebsiella pneumoniae* ESBL (>100.000 CFU/mL), possibly due to prolonged catheterization. The pathogen was sensitive to ciprofloxacin, meropenem, gentamycin, and amikacin (Table 4).

Concluding from the clinical presentation, the patient was initially diagnosed with brief psychotic disorder and catatonic disorder due to another medical condition (according to DSM-5-TR) or secondary catatonia syndrome (according to ICD-11) [12,30].

### 2.4. Treatment and Clinical Evolution

Due to a shortage of injectable lorazepam, the initial catatonia treatment was based on a dual protocol with lorazepam and diazepam, previously reported in the literature [31]. One ampule of lorazepam (4 mg/mL) was intravenously administered, then a second one was administered 2 h later, after no response was observed. Then, diazepam was infused continuously (10 mg/500 mL in normal saline at a rate of 1.25 mg/h or 62.5 mL/h, totaling 30 mg/day) and constantly monitored for respiratory depression, until catatonic symptoms improved.

The initial psychotropic treatment was stopped and the patient was switched to quetiapine as an antipsychotic, slowly titrated to 800 mg/day p.o. After catatonic symptoms partially relieved, diazepam was slowly tapered off and replaced with oral clonazepam, initially titrated up to 1.5 mg/day p.o., then gradually reduced to 0.5 mg/day p.o.. Propranolol (40 mg/day p.o.) was additionally added to alleviate the superior limbs tremor. It was administered in lower doses than usual, due to transient hypotension and bradycardia.

Additional medications included analgesics and antipyretics (acetaminophen 1000 mg/day i.v. and dipyrone 1000 mg/day i.v.), used during fever and headaches, vitamins (combination of thiamine and pyridoxine 100/50 mg/day i.v., ascorbic acid 1000 mg/day i.v., both stopped before discharge, and folic acid 5 mg/day p.o.), hepatoprotective (silymarin 300 mg/day p.o., stopped before discharge), laxatives (glycerin suppositories 1/day and lactulose 30 mL/day p.o., both stopped after bowel transit resumed), enemas, and enteral nutrition.

The treatment for the *K. pneumoniae* ESBL UTI consisted of ciprofloxacin 1000 mg/day p.o., 10 days, and *Saccharomyces boulardii* probiotic, 10 days. Both were prescribed for outpatient treatment.

After 17 days of hospitalization, the patient was discharged with improved symptoms. The final treatment regimen consisted of quetiapine 800 mg/day p.o., clonazepam 0.5 mg/day p.o., propranolol 40 mg/day p.o., folic acid 5 mg/day p.o., ciprofloxacin 1000 mg/day p.o. (10 days), and *S. boulardii* probiotic (10 days).

### 2.5. Timeline

A timeline of the evolution of the psychotropic treatment during the admission to the Baia Mare County Emergency Hospital was previously presented in Figure 1.

A timeline of the psychotropic medication changes and investigations conducted during the admission to the Cluj-Napoca County Emergency Hospital is illustrated in Figure 3.

### 2.6. Follow-Up and Outcomes

The patient presented for follow-up one month after discharge with significant improvements in symptoms. Clonazepam was gradually tapered off consequently. During a new follow-up, three months later after the discharge, the patient seemed fully functional and resumed his daily activities. No signs of tardive catatonia or psychosis relapse were observed.

## 3. Discussion

In this clinical case, we attempted to establish the etiology of the catatonia in this patient, by ruling out infectious, vascular, neurological, and endocrine causes [32,33]. However, all performed investigations could not point out any of the organic causes taken into consideration. Therefore, the diagnosis was not clear by the time the patient was discharged. After performing a medication review and reanalyzing the routine blood test results, a new hypothesis emerged in this case.

We believe that the patient suffered a moderate form of NMS, characterized by moderate muscular rigidity, confusion or catatonia, temperatures between 38 and 40 °C, and heart rates between 100 and 120 bpm, according to the classification provided by the DGPPN S3 guideline for schizophrenia [20,21,22,23]. Differential diagnosis between NMS and catatonia was conducted according to the DSM-5-TR criteria for both nosological entities (Table 8) [12].

On first sight, the patient presented criteria for both NMS and catatonia. According to the DSM-5-TR diagnosis criteria for NMS, the patient had been exposed to a dopamine antagonist within 72 h prior to the symptoms’ development (as depicted in Figure 1) and presented hyperthermia. Abnormal laboratory test results included elevated creatine kinase levels at least four times over the normal range and leukocytosis (although this can be attributed to the bronchitis and later to the UTI), while the CSF findings and neuroimaging were normal. Additionally, the EEG depicted generalized slowing [12].

The diagnosis of catatonia in this patient was supported by the presence of stupor, catalepsy, waxy flexibility, mutism, posturing, grimacing, and echolalia, with a score of 30 out of 69 points on the BFCRS, representing a catatonia of moderate severity [12,26]. Additionally, prodromal symptoms that debuted 2 weeks prior to the admission were observed [18].

Our argument for the hypothesis of NMS diagnosis is represented by the psychotropic medication administered by the patient prior to the admission, with respect to haloperidol, aripiprazole, and valproic acid. Haloperidol is one of the most often typical antipsychotics implicated in NMS, along with fluphenazine and pimozide [34]. There have been literature reports regarding NMS induced by low doses of aripiprazole in adolescents [35,36], while valproic acid is specifically mentioned in the DSM-5-TR as being capable of inducing NMS [12]. Additionally, reports regarding valproic acid-induced NMS in adolescents are available in the literature [37]. By applying the Adverse Drug Reaction (ADR) Probability Score, developed by Naranjo et al. [38], we obtained a total score of 4 points, meaning a possible correlation between the administered psychotropic treatment and the NMS symptoms in this case.

To strengthen the diagnosis of NMS in this case report, we applied two different instruments, namely the NMS Diagnostic Criteria [39] and Levenson’s criteria [40]. According to the NMS Diagnostic Criteria, we obtained a total score of 85 out of 100 points, which was above the threshold of 74 points needed to confirm the NMS diagnosis [39]. Additionally, according to Levenson’s criteria, this patient presented all three major manifestations listed in the criteria (fever, rigidity, and elevated creatine kinase levels—Levenson’s triad), thus indicating a high probability of the presence of NMS in this case [40]. Levenson’s criteria do not differentiate between the types of rigidity, “lead pipe” or waxy.

Regarding the new psychotropic treatment, quetiapine administration was motivated by the presence of paranoid delusions in the thought content. While the hypothesis of NMS was not initially brought up during the admission, we believe that quetiapine was an adequate choice even in the context of an NMS. Both quetiapine and clozapine are the first-choice antipsychotics after the resolution of NMS symptoms. Some sources mention that antipsychotic use during catatonia is not advised, due to unclear etiology (could be either NMS or malignant catatonia or none of them). However, there is evidence that supports concomitant administration of the antipsychotic with a benzodiazepine, which was also performed in this clinical case [11,41].

Based on these discussions, we consider that the final diagnosis in this case should be moderate NMS. According to the DSM-5-TR, this diagnosis invalidates the initial diagnosis of catatonia disorder due to another medical condition, as it should not have been established in the context of NMS [12].

## 4. Limitations

Several limitations were identified in the management of this case. The first limitation was represented by the single-case design of the current report. The presented case represents a rare occurrence, especially in our service. Therefore, a case series design would have been difficult in this situation and would have prolonged the dissemination of the results. The single-case design leads to limited generalization of the reported findings, thus representing the second identified limitation.

The third identified limitation of the case report was represented by the diagnostic uncertainty. While extensive discussions were made regarding the differential diagnosis between catatonia and NMS, the proposed diagnosis of NMS lacked some key features, which were identified as vulnerabilities: the borderline values for the abnormal vital signs and the absence of “lead pipe” rigidity. Additionally, several investigations and tests were not performed, which could have strengthened the diagnosis of NMS, such as dopamine agonist challenge, confirmatory muscle biopsy, serum iron, and urine myoglobin. The potential confounding infections and medications could also have affected the strength of the diagnosis.

The fourth identified limitation was represented by the selected treatment options. While electroconvulsive therapy (ECT) should have been initiated after the first failed trial with benzodiazepines [6], it is important to note that ECT was not available in our service, but only in two other centers from Romania. A transfer to one of those centers in an effective manner would have been difficult in the given conditions. Another vulnerability regarding the treatment was represented by the choice to initiate quetiapine before complete recovery of the patient. While this contradicts the established guidelines [6], we found evidence in other sources that supports the concomitant administration of antipsychotics with benzodiazepines [41].

The fifth identified limitation was represented by the potential inadequate monitoring through laboratory tests. While day-by-day monitoring is essential in NMS and rhabdomyolysis, this was difficult to provide due to limited available resources.

A sixth minor limitation of the current report was that the BFCRS and PANSS scales were performed by non-certified raters, thus possibly altering the obtained scores.

## 5. Conclusions

In conclusion, this report presented the clinical case of a patient with moderate NMS associated with catatonia, following exposure to haloperidol, valproic acid, and aripiprazole. Initially thought to be a case of catatonic disorder due to another medical condition, all considered possible organic causes (infectious, vascular, endocrine, and neurological) were ruled out. The diagnosis of NMS was confirmed according to the DSM-5-TR and DGPPN S3 guidelines for schizophrenia and strengthened by applying the NMS Diagnostic Criteria Score and Levenson’s criteria. The new diagnosis invalidated the previous one of catatonic disorder due to another medical condition, according to the DSM-5-TR.

The case highlighted the thin line between catatonia and NMS and the possibility of concomitantly manifesting in the same patient. The case may have potential to serve as a reference for future cases of NMS associated with catatonia, considering the ongoing difficulties in establishing diagnoses involving both entities.

## Figures and Tables

**Figure 1 reports-08-00134-f001:**
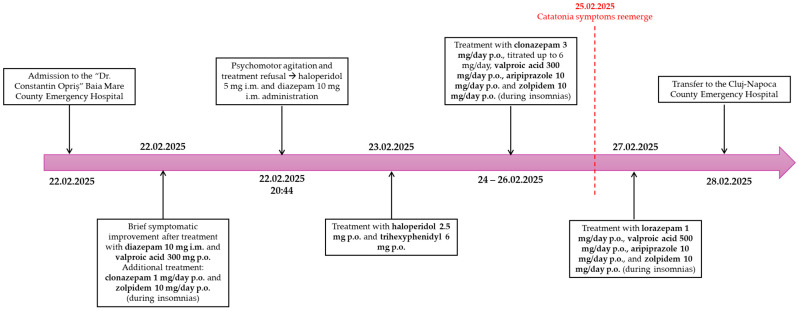
Timeline of the evolution of the psychotropic treatment during the admission to the Baia Mare County Emergency Hospital.

**Figure 2 reports-08-00134-f002:**
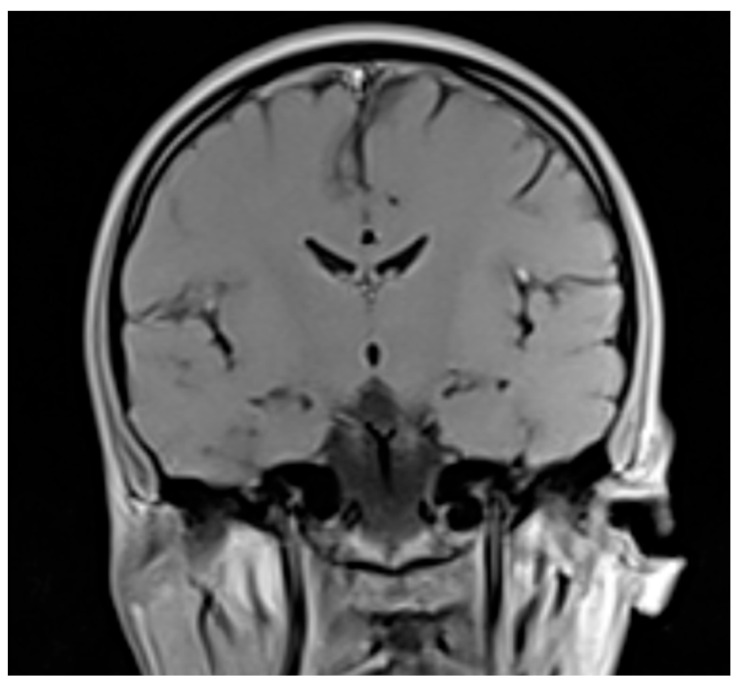
Cerebral MRI T1 with contrast, coronal section, without pathological changes (performed on 28 February 2025).

**Figure 3 reports-08-00134-f003:**
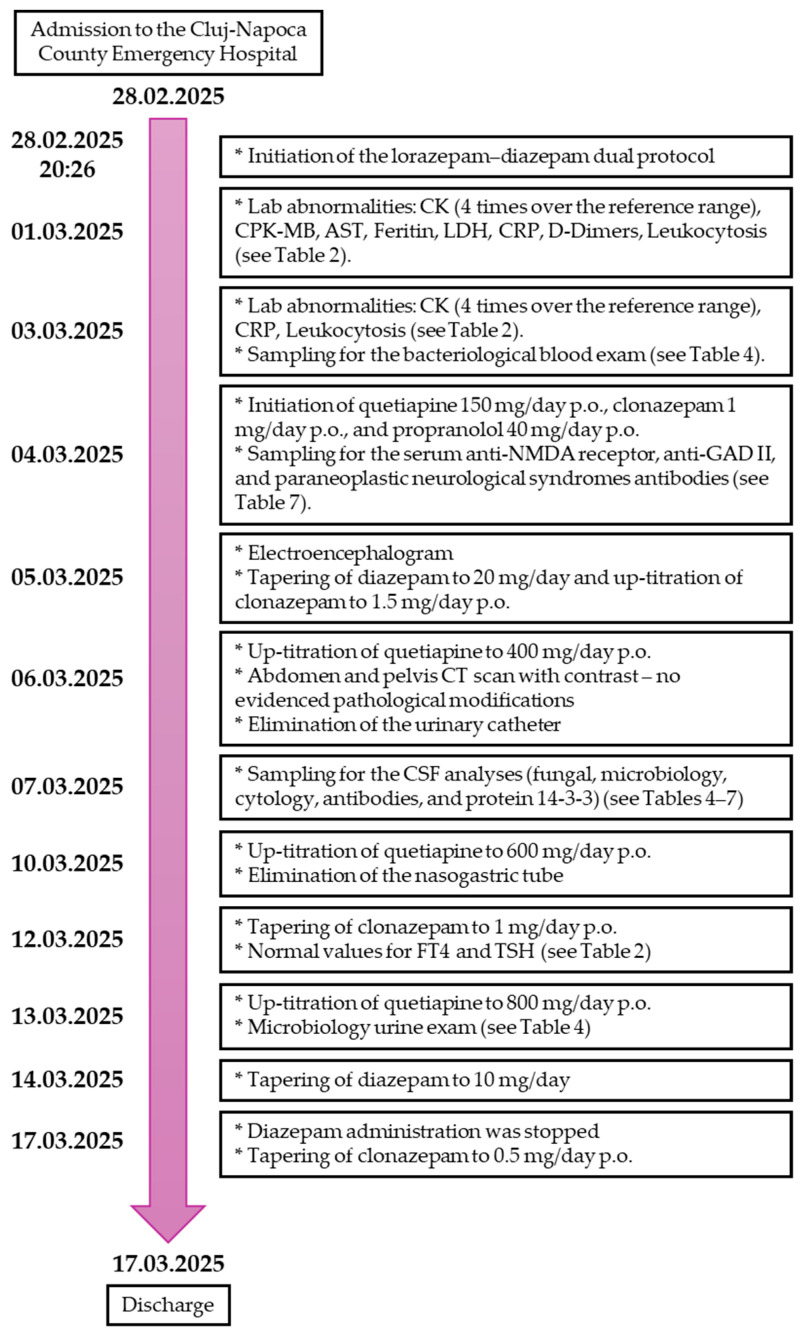
Timeline of the psychotropic medication changes and investigations conducted during the admission to the Cluj-Napoca County Emergency Hospital.

**Table 1 reports-08-00134-t001:** Clinical stages of NMS and associated clinical presentation.

NMS Stage	Clinical Symptoms	Reference
Stage I (Drug-induced Parkinsonism)	Rigidity, tremor	[20,21,22,23]
Stage II (Drug-induced Catatonia)	Rigidity, tremor, stupor	[20,21,22,23]
Stage III (Mild NMS)	Mild rigidity, catatonia or confusion, temperature ≤ 38 °C, heart rate ≤ 100 bpm	[20,21,22,23]
Stage IV (Moderate NMS)	Moderate rigidity, catatonia or confusion, temperature 38–40 °C, heart rate 100–120 bpm	[20,21,22,23]
Stage V (Severe NMS)	Severe rigidity, catatonia or confusion, temperature ≥ 40 °C, heart rate ≥ 120 bpm	[20,21,22,23]

**Table 2 reports-08-00134-t002:** Summary of the abnormal values of routine tests.

		1 March 2025		3 March 2025		5 March 2025		7–12 March 2025	
Parameter	Reference Range	Value	Interpretation	Value	Interpretation	Value	Interpretation	Value	Interpretation
ALT	<50 U/L	49 U/L	✓	50 U/L	✓	-	-	-	-
AST	<50 U/L	159 U/L	↑	71 U/L	↑	48 U/L	-	-	-
CK	<171 U/L	4908 U/L	↑	1270 U/L	↑	393 U/L	↑	-	-
CPK-MB	<24 U/L	48 U/L	↑	-	-	-	-	-	-
Feritin	20–250 ng/mL	320 ng/mL	↑	-	-	-	-	-	-
Glycemia	74–106 mg/dL	67 mg/dL	↓	-	-	-	-	73 mg/dL	↓
LDH	<248 U/L	438 U/L	↑	-	-	-	-	302 U/L	↑
CRP	<0.5 mg/dL	6.27 mg/dL	↑	1.80 mg/dL	↑	2.01 mg/dL	↑	-	-
D-Dimers	<243 ng/mL	12,189 ng/mL	↑	-	-	-	-	660 ng/mL	↑
WBC	4.5–11.5 × 10^9^/L	13.20 × 10^9^/L	↑	13.49 × 10^9^/L	↑	13.43 × 10^9^/L	↑	15.17 × 10^9^/L	↑
NEUT#	1.5–6.6 × 10^9^/L	10.45 × 10^9^/L	↑	12.09 × 10^9^/L	↑	11.27 × 10^9^/L	↑	11.84 × 10^9^/L	↑
MONO#	0.21–0.92 × 10^9^/L	1.03 × 10^9^/L	↑	0.58 × 10^9^/L	✓	0.87 × 10^9^/L	✓	0.93 × 10^9^/L	↑
ESR	<15 mm/h	-	↑	71 mm/h	-	-	-	-	-
Folic acid	5.9–23.2 ng/mL	-	-	-	-	-	-	2.83 ng/mL	↓
B12 vitamin	180–914 pg/mL	-	-	-	-	-	-	329 pg/mL	✓
FT4	0.9–1.53 ng/dL	-	-	-	-	-	-	1.29 ng/dL	✓
TSH	0.51–4.17 uIU/mL	-	-	-	-	-	-	1.55 uIU/mL	✓

Legend: ALT—alanine aminotransferase; AST—aspartate aminotransferase; CK—creatine kinase; CPK-MB—creatine phosphokinase-MB; LDH—lactate dehydrogenase; CRP—C-reactive protein; WBC—white blood count; NEUT#—neutrophil count; MONO#—monocyte count; ESR—erythrocyte sedimentation rate; FT4—free thyroxine; TSH—thyroid-stimulating hormone; uIU—millionth of an International Unit; ✓—value in the normal range of the parameter; ↑—value above the normal range of the parameter; ↓—value below the normal range of the parameter.

**Table 3 reports-08-00134-t003:** Summary of the observations from the native cranial CT scan and cerebral MRI scan with contrast.

Date	Test	Result
**26 February 2025**	Native cranial CT scan	No cerebral oedema. No suggestive intracerebral heterodense lesions for increased intracranial pressure of any cause or recent ischemic stroke. No cerebral parenchymatous hemorrhagic zones. Ventricular system on the median line, with normal content. Free subarachnoid spaces.
**28 February 2025**	Cerebral MRI scan with contrast	No areas of acute ischemia. No intra- or extracerebral hemorrhagic accumulations. No evidenced lesions on the cerebral parenchyma, supra- or infratentorial. No evidenced cerebral or meningeal pathological modifications by the contrast.

**Table 4 reports-08-00134-t004:** Summary of the microbiology examinations (bacterial and fungal) performed on blood, urine, and CSF samples.

Date	Test	Result	Observations
**3 March 2025**	Bacteriological blood exam	No developed aerobic and/or anaerobic bacteria after 5 days of incubation	-
**7 March 2025**	Fungal CSF exam	Absent *Candida* spp.	-
**7 March 2025**	Microbiology CSF exam	Absent *Streptococcus* spp., *Staphylococcus aureus*, Enterobacterales, *Enterococcus* spp., *Pseudomonas* spp., *Acinetobacter* spp., *Candida* spp.	-
**7 March 2025**	CSF cytology	Acellular smears	-
**13 March 2025**	Microbiology urine exam	*Klebsiella pneumoniae* ESBL 100.000 CFU/mL	Sensitive to meropenem, gentamycin, ciprofloxacin, and amikacin. Resistant to cefuroxime axetil, ceftazidime, amoxicillin + clavulanic acid, and ceftriaxone

Legend: CSF—cerebrospinal fluid; ESBL—extended-spectrum beta-lactamase; CFU—colony-forming unit.

**Table 5 reports-08-00134-t005:** Summary of the serum immunology examinations for possible viral infections (performed on 7 March 2025).

Test	Result	Reference Range
Anti-Herpes simplex type 1 IgG antibodies	Non-reactive	Non-reactive
Anti-Herpes simplex type 1 IgM antibodies	Non-reactive	Non-reactive
Anti-Herpes simplex type 2 IgG antibodies	Non-reactive	Non-reactive
Anti-Herpes simplex type 2 IgM antibodies	Non-reactive	Non-reactive
Anti-Cytomegalovirus IgM antibodies	Non-reactive	Non-reactive
Anti-Cytomegalovirus IgG antibodies	<4 AU/mL	<4 AU/mL—Non-reactive ≥4–<6 AU/mL—Uncertain ≥6 AU/mL—Reactive
Anti-VZV IgG antibodies	Reactive	Non-reactive
Anti-VZV IgM antibodies	Non-reactive	Non-reactive
Anti-Epstein–Barr virus VCA IgM antibodies	Non-reactive	Non-reactive

Legend: IgG—immunoglobulin G; IgM—immunoglobulin M; VZV—Varicella zoster virus; VCA—viral capsid antigen; AU—arbitrary unit.

**Table 6 reports-08-00134-t006:** Results of the multiplex CSF panel (performed on 7 March 2025).

Pathogen	Result *
Enterovirus	Undetectable
Herpes simplex virus 1	Undetectable
Herpes simplex virus 2	Undetectable
Human parechovirus	Undetectable
Human herpes virus 6	Undetectable
Varicella zoster virus	Undetectable
*S. pneumoniae*	Undetectable
*Neisseria meningitidis*	Undetectable
*S. agalactiae*	Undetectable
*Listeria monocytogenes*	Undetectable
*Hemophilus influenzae*	Undetectable
*Escherichia coli* K1	Undetectable
*S. pyogenes*	Undetectable
*Mycoplasma pneumoniae*	Undetectable
*Cryptococcus neoformans/gattii*	Undetectable

*—Laboratory note regarding the results: the obtained result is influenced by sampling technique, storage, and transport of the samples. The results interpretation is based on the clinical and epidemiological data of the patient. Prior to the sampling, no intrathecal substances should be administered. A negative result does not exclude an infection. A positive result for herpes simplex virus (HSV) and Varicella zoster virus (VZV) could be attributed to a primary or latent infection. The result may be false-negative in the first 4 days since the debut of the HSV infection. A positive test for human herpes virus 6 (HHV-6) does not differentiate between primary, self-limiting, and reactive infections and chromosomally integrated HHV-6. HHV-6 may be false-negative in hemorrhagic CSF. The test detects only the encapsulated strains of *N. meningitidis* and *E. coli* strains with K1 capsular antigen. The result for Enterovirus may be subjected to cross-reactivity with human rhinoviruses.

**Table 7 reports-08-00134-t007:** Summary of the serum and CSF analyses for the detection of anti-NDMA receptor encephalitis, prion diseases, and paraneoplastic neurological syndromes. The analyses were performed as follows: 4 March 2025—all serum analyses; 7 March 2025—all CSF analyses.

Test	Sample Type	Result	Reference Range
Anti-NMDA receptor antibodies	CSF	<1:1	<1:1
	Serum	<1:10	<1:10
Anti-GAD II antibodies	CSF	negative	-
	Serum	<5.00 UI/mL	<10.00 UI/mL—negative ≥10.00 UI/mL—positive
Anti-Amphiphysin IgG antibodies	CSF	negative	negative
	Serum		
Anti-C2 IgG antibodies	CSF	negative	negative
	Serum		
Anti-PNMA2 (Ma2/Ta) IgG antibodies	CSF	negative	negative
	Serum		
Anti-Ri IgG antibodies	CSF	negative	negative
	Serum		
Anti-Hu IgG antibodies	CSF	negative	negative
	Serum		
Anti-Yo IgG antibodies	CSF	negative	negative
	Serum		
Anti-Recoverin IgG antibodies	CSF	negative	negative
	Serum		
Anti-SOX1 IgG antibodies	CSF	negative	negative
	Serum		
Anti-Titin IgG antibodies	CSF	negative	negative
	Serum		
Anti-GAD64/65 antibodies	CSF	negative	negative
Anti-Tr/DNER antibodies	CSF	negative	negative
Protein 14-3-3	CSF	negative *	-

Legend: NMDA—*N*-methyl-*D*-aspartate; GAD II—glutamic acid decarboxylase II; IgG—immunoglobulin G; C2—component C2; PNMA2—human paraneoplastic antigen Ma2; SOX1—SRY-related high-mobility group box 1; GAD64/65—glutamic acid decarboxylase 64/65; Tr/DNER—Trotter antigen/Delta/Notch-like epidermal growth factor-related receptor. *—Laboratory note regarding the protein 14-3-3 result: the proteins 14-3-3 were not detectable as elevated in the CSF. With the currently available laboratory chemistry methods, no evidence of prion disease could be found. However, due to the negative CSF findings, a prion disease cannot be excluded. Negative results can be obtained in genetic prion diseases (e.g., lethal familial insomnia, Gertsmann–Sträussler–Scheinker syndrome).

**Table 8 reports-08-00134-t008:** Differential diagnosis between NMS and catatonia according to the DSM-5-TR criteria for both nosological entities [12].

NMS	Catatonia **
Exposure to a dopamine antagonist within 72 h prior to the symptom development	Stupor
Hyperthermia (>38.0 °C on at least two occasions, orally measured) with profuse diaphoresis *	Catalepsy
Generalized rigidity, described as “lead pipe” rigidity	Waxy flexibility
CK levels elevated at least 4 times over the reference range	Mutism
Changes in mental status (delirium or altered consciousness)	Negativism
Autonomic activation and instability—tachycardia (>25% above the baseline values), diaphoresis, elevated blood pressure (≥20 mmHg for the diastolic blood pressure or ≥25 mmHg for the systolic blood pressure within 24 h)	Posturing
Urinary incontinence	Mannerism
Pallor	Stereotypy
Tachypnea (>50% above the baseline values)	Agitation, not influenced by external stimuli
Respiratory distress	Grimacing
Laboratory abnormalities (leukocytosis, metabolic acidosis, hypoxia, decreased serum iron levels, increased serum muscle enzymes levels and catecholamines)	Echolalia
	Echopraxia
	±Evidence that the disturbance is the direct pathophysiological consequence of another condition (for catatonic disorder due to another medical condition)

Legend: *—hyperthermia with profuse diaphoresis is considered a distinguishing feature for the NMS; **—three or more of the twelve listed features of catatonia are necessary to confirm the diagnosis [12].

## Data Availability

The original contributions presented in this study are included in the article. Further inquiries can be directed to the corresponding author.

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
