# Peer review of "Catatonia in a Possible Case of Moderate Neuroleptic Malignant Syndrome: A Case Report"

_reports, 2025, doi:10.3390/reports8030134_

Round 1
Reviewer 1 Report
Comments and Suggestions for Authors
The issue of NMS criteria is focus of concern. If you apply DSM-5-TR criteria to this patient, then the patient meets criteria. Alternately, if you had another patient with malignant catatonia that patient would meet DSM-5-TR criteria. This may be because DSM-5-TR is designed to detect cases of NMS, even atypical NMS.
If the authors wish to apply criteria. Consider:
Gurrera R, Caroff SN, Cohen A, Carroll BT, DeRoos F, Francis A, Frucht S, Gupta S, Levenson JL, Mahmood A, Mann SC, Policastro MA, Rosebush PI, Rosenberg H, Sachdev PS, Trollor JN, Velamoor VR, Watson CB, Wilkinson JR. An International Consensus Study of Neuroleptic Malignant Syndrome Diagnostic Criteria Using the Delphi Method. J of Clinical Psychiatry 72 (9), 1222-1228, 2011
So the criteria may be less restrictive.
There is an earlier effort to look at the overlap of NMS and Malignant catatonia:
Carroll BT, Taylor RE. The Nondichotomy Between Lethal Catatonia and Neuroleptic Malignant Syndrome. Journal of Clinical Psychopharmacology 17(3):235-236, 1997
See also: Lee JWY, Neuroleptic-induced catatonia
10.1097/JCP.0b013e3181c9bfe6
I agree with the conclusion.
Also, the EEG slowing is consistent with NMS or Catatonia due to a general medical condition.
Reviewer 2 Report
Comments and Suggestions for Authors
Manuscript ID: reports-3748498
In this paper, the authors report a case of catatonic syndrome that developed in the context of moderate neuroleptic malignant syndrome in a patient with a recent history of taking antipsychotics. Although the topic itself is interesting, in my opinion the paper requires a few minor modifications.
Abstract section
The abstract covers the main aspect of the report.
Some keywords need to be corrected. Certain terms (e.g. autoimmune encephalitis) are not included in the text of the abstract.
Introduction and Clinical Significance sections
The introduction should include more details about the relationship of drugs to neuroleptic malignant syndrome and catatonia, including the possible risk of drug-drug interactions.
The name and appropriate acronym (e.g. DGPPN S3) must be written in full when first described in the text, then only the acronym is used (e.g. lines 66).
In line 68, the appropriate reference should be inserted that refers to the DGPPN S3 guideline for schizophrenia.
Case Presentation sections
The case presentation is clear, however,
References 14-17 should also be included in the appropriate section of Table 1, not in the title of Table 1.
Table 3 needs to be rearranged in the Result and Observations section to make it more understandable for readers.
Discussion section
A discussion of the limitations of the study is missing in the Discussion section.
Conclusion section
The conclusions are valid, but too general. The Conclusions section needs to be improved in terms of introducing order into the writing using the analyzed variables, causes, consequences, and future benefits of this report.

Reviewer 3 Report
Comments and Suggestions for Authors
General Assessment:
This case report presents an interesting and clinically relevant diagnostic challenge, describing the overlap between catatonia and a possible moderate neuroleptic malignant syndrome (NMS) in an adolescent patient. The authors provide a well-structured clinical evolution, extensive diagnostic work-up, and treatment response. The manuscript is mostly well-written, and the topic is of value to clinicians working in psychiatry and related fields.
- Does the introduction provide sufficient background and include all relevant references?
The introduction provides a general overview of catatonia and NMS, referencing DSM-5-TR, ICD-11, and relevant guidelines. However, the background could be strengthened by better emphasizing the diagnostic difficulty in differentiating between NMS and catatonia, particularly in young patients. Additionally, a clearer summary of recent epidemiological data or diagnostic frameworks would provide stronger context for the case’s relevance. Some important literature is referenced, but the integration into the argument could be more cohesive.
- Is the research design appropriate?
For a case report, the structure and progression of the narrative are appropriate. The clinical course is well-documented, and the diagnostic reasoning is logically developed. The retrospective re-evaluation of the case, suggesting a likely NMS diagnosis, is handled thoughtfully and adds value. While retrospective, the approach is consistent with standards for case-based evidence.
- Are the methods adequately described?
The manuscript offers detailed and transparent reporting of medications, psychiatric and physical evaluations, lab results, imaging, and ancillary diagnostics. The sequence of interventions is clearly outlined. The psychometric tools used are standard and appropriately cited. Minor clarifications, such as precise timelines or clinical thresholds, could enhance clarity but are not critical.
- Are the results presented?
Clinical findings are systematically reported and supported by comprehensive tables. The narrative is easy to follow, and the grouping of data by investigation type (lab, imaging, serology, CSF, EEG) is logical and well-organized. The inclusion of psychometric scores further enriches the clinical picture. The tables are clear, informative, and appropriately referenced within the text.
- Are the conclusions supported by the results?
The proposed diagnosis of moderate NMS is plausible, especially in light of the medication history and elevated creatine kinase. However, the authors should be more explicit in aligning the clinical data with standardized diagnostic criteria for NMS (e.g., Levenson’s criteria or DSM-5-TR). Some of the vital sign abnormalities were borderline, and it would be beneficial to acknowledge this uncertainty more explicitly. While the conclusion is interesting and relevant, its evidentiary basis could be strengthened with a clearer link to diagnostic criteria.
- Are all figures and tables clear and well-presented?
Tables are clean, concise, and relevant. Laboratory results are presented with both reference ranges and their corresponding interpretations. The MRI figure is of limited added value (as it is negative), but its inclusion is acceptable. Overall, visual data presentation supports the narrative effectively.
Final Recommendation: Minor revisions
The manuscript offers valuable insights into the diagnostic complexity of catatonia and NMS in adolescents. With minor revisions — particularly expanding the introduction, explicitly linking findings to diagnostic criteria, and clarifying the retrospective diagnostic reasoning — this case report would be suitable for publication. The topic is timely, educational, and clinically important.
Comments on the Quality of English LanguageQuality of the English Language
The manuscript is generally understandable, but there are numerous instances of awkward phrasing, unnatural word order, and grammatical inaccuracies that may hinder clarity. The frequent use of repetitive structures (e.g., "conditions, conditions"), literal translations (e.g., “paraclinical investigations”), and overly long or indirect sentences affects the overall fluency and academic tone. For example, phrases such as “No drinking or substance consumption history was identified” could be revised to “There was no history of alcohol or illicit substance use” for better clarity and correctness. Similarly, the phrase “treatment of the catatonia syndrome in our service” would benefit from refinement to “catatonia treatment was initiated using a dual lorazepam-diazepam protocol.”
A thorough English language edit by a native speaker or professional editor is recommended to improve flow, grammar, and style throughout the manuscript.
Reviewer 4 Report
Comments and Suggestions for Authors
Manuscript reviewed: Catatonia in a Possible Case of Moderate Neuroleptic Malignant Syndrome: A Case Study Report
The authors present an 18‑year‑old male who developed a two‑week history of stuporous catatonia with waxy flexibility, hypertonia and psychotic symptoms. Routine work‑up ruled out infectious, neurological and endocrine causes. Marked creatine‑kinase elevation, low‑grade fever (38.2 °C) and prior exposure to haloperidol, aripiprazole and valproic acid led the team to post‑hoc label the picture as “possible moderate neuroleptic malignant syndrome (NMS) with catatonic features”. A benzodiazepine protocol (IV lorazepam bolus followed by continuous diazepam) produced gradual clinical improvement; quetiapine was introduced once rigidity abated. The patient was discharged after 17 days and remained improved at one‑month follow‑up.
The report offers a well‑structured chronology with extensive laboratory documentation and appropriate references contemporary catatonia/NMS guidelines, making it a potentially useful teaching vignette for trainees.
Some issues should be considered:
Major issues
- Unresolved diagnostic ambiguity. The manuscript concludes that the episode represents “moderate NMS”, yet cardinal NMS criteria (sustained hyperthermia > 38.5 °C, “lead‑pipe” rigidity, autonomic instability) were only partially met and the BFCRS score was compatible with primary catatonia. A differential table contrasting DSM‑5‑TR criteria for catatonia vs. NMS should be provided and the final diagnosis justified more rigorously (e.g., Levenson’s triad).
- No standardized NMS severity or causality instrument. Scales such as the NMS Diagnostic Criteria Score (Mann et al., 2011) or the Naranjo algorithm were not applied; thus, the strength of the NMS attribution is weak.
- The timeline of antipsychotic exposure is unclear. Doses and exact discontinuation times for haloperidol, aripiprazole and valproate relative to symptom onset are not stated, preventing an assessment of the 72‑hour exposure window stipulated by DSM‑5‑TR.
- Therapeutic strategy may conflict with guidelines. High‑dose quetiapine (up‑titrated to 800 mg/day) was started before full resolution of catatonia/NMS; current British Association for Psychopharmacology guidance recommends withholding antipsychotics until complete recovery and normalization of CK; Rogers et al., 2023.
- Laboratory trend data insufficient. Only peak CK, CRP and leukocyte values are given; day‑by‑day trends (essential for monitoring NMS or rhabdomyolysis) are missing, as are serum iron or urine myoglobin.
- Missing imaging and CSF details. The text states that CT/MRI and CSF were “normal”, but exact sequences, timing and reference ranges are not supplied, limiting reproducibility.
- Outcome assessment too short. A single one‑month follow‑up is reported; functional recovery, relapse of psychosis or tardive catatonia at ≥6 months should be documented.
- Absence of a dedicated ‘Limitations’ section. The Discussion omits a critical appraisal of single‑case design, diagnostic uncertainty, treatment confounders and limited generalizability―contravening CARE/Case‑Report guidelines.
- Ethics statement insufficient. Although informed consent is declared, the manuscript should clarify whether the patient (with moderate intellectual disability) had decision capacity and whether consent was additionally obtained from a legal guardian.
- Literature contextualization is narrow. Key systematic reviews on catatonia treatment (e.g., Cuevas‑Esteban et al., 2021; Sienaert et al., 2020) are not cited, and the discussion overlooks ECT as a second‑line option when benzodiazepines fail.
Minor issues
- Title formatting. “Case Study Report” is redundant; consider “Case Report” per journal style.
- Abstract length exceeds journal limit (≈250 words). Reduce repetitive background lines and include objective numerical data (BFCRS = 30, peak CK = 1 450 U/L).
- Introduction paragraphs repeat historical definitions (lines 41‑48, 52‑60) without adding new insight; condense and cite seminal reviews (Fink & Taylor, 2003).
- Table 1 (NMS stages) lacks citation superscripts and normal ranges.
- Table 2 (lab abnormalities) omits reference intervals and units for each parameter, impeding clinical interpretation.
- Psychometric scales. Provide version/language of BFCRS and PANSS, specify whether ratings were performed by certified raters, and explain clinical meaning of a PANSS total = 127.
- Medication section. Clarify whether diazepam infusion rate (1.25 mg h‑1) was monitored for respiratory depression; report serum benzodiazepine levels if measured.
- Grammar and style. Several sentences are lengthy or contain minor errors (e.g., “was communicating only with his parents” line 90). A professional language edit is recommended.
- Reference accuracy. Check citation [9] “Ungurean D.” appears incomplete; URL citations (UpToDate) should include accessed date and be limited.
- Self‑citation rate. Six of 30 references are by the same author group; ensure balance with external literature.
- Abbreviations list duplicates items already defined in text (e.g., CK, CRP) and includes laboratory acronyms not used in the manuscript (e.g., HIV, VZV).
- Figure or flow‑chart. A timeline figure illustrating symptom onset, medication changes, CK trend and vital signs would greatly enhance readability.
Missing “Limitations” section (required)
I strongly recommend adding a dedicated section that transparently discusses single‑case design, lack of confirmatory muscle biopsy or dopamine‑agonist challenge, short follow‑up, and potential confounding from concurrent infections and medications.
Suggested key references
Cuevas‑Esteban, J., et al. (2021). Benzodiazepines versus electroconvulsive therapy for catatonia: A systematic review and meta‑analysis. Journal of Affective Disorders, 292, 476–484.
Fink, M., & Taylor, M. A. (2003). Catatonia: A clinician’s guide to diagnosis and treatment. Cambridge University Press.
Rogers, J. P., Oldham, M. A., Fricchione, G., et al. (2023). Evidence‑based consensus guidelines for the management of catatonia. Journal of Psychopharmacology, 37, 327–369.
Sienaert, P., Rooseleer, J., & De Fruyt, J. (2020). ECT in the treatment of catatonia: A systematic review. Acta Psychiatrica Scandinavica, 141, 308‑321.
Round 2
Reviewer 4 Report
Comments and Suggestions for Authors
Authors have mainly addressed the reviewer's comments.